# When Fairness Meets Privacy: Fair Classification with Semi-Private Sensitive Attributes

**Canyu Chen[1], Yueqing Liang[1], Xiongxiao Xu[1], Shangyu Xie[1], Yuan Hong[2], Kai Shu[1]**
[1] Department of Computer Science, Illinois Institute of Technology USA
[2] Department of Computer Science & Engineering, University of Connecticut USA
{cchen151, yliang40, xxu85, sxie14}@hawk.iit.edu, yuan.hong@uconn.edu, kshu@iit.edu

## Abstract

Machine learning models have demonstrated promising performances in many areas. However, the concerns that they can be biased against specific groups hinder their adoption in high-stake applications. Thus, it is essential to ensure fairness in machine learning models. Most of the previous efforts require access to sensitive attributes for mitigating bias. Nevertheless, it is often infeasible to obtain a large scale of data with sensitive attributes due to people's increasing awareness of privacy and the legal compliance. Therefore, an important research question is *how to make fair predictions under privacy*. In this paper, we study a novel problem of *fair classification in a semi-private* setting, where most of the sensitive attributes are private and only a small amount of clean ones are available. To this end, we propose a novel framework FAIRSP that can first *learn to correct* the noisy sensitive attributes under the privacy guarantee by exploiting the limited clean ones. Then, it jointly models the corrected and clean data in an adversarial way for debiasing and prediction. Theoretical analysis shows that the proposed model can ensure fairness when most sensitive attributes are private. Extensive experimental results in real-world datasets demonstrate the effectiveness of the proposed model for making fair predictions under privacy and maintaining high accuracy.

## 1 Introduction

Machine learning has shown promising performances in various high-stake applications such as disease identification [40], crime prediction [39], and loan application filtering [21]. However, in these applications, an emerging concern is that the prediction derived from machine learning models can often be biased and unfair to specific (and often marginalized) groups. For example, racial bias in medical analysis can lead to disparate treatments [40]. In addition, a recent Forbes report shows that machine learning bias has caused 80% of the black mortgage applicants to be denied[1]. Such discrimination can have detrimental societal effects that weaken the public trust among individuals, groups and the society. Therefore, it is critical to ensure fairness in machine learning for social good.

The recent advancements of fair machine learning–aiming to develop effective algorithms to achieve fairness and maintain good prediction performance–have attracted increasing attention [36, 27]. The majority of existing fair machine learning models require direct access to *sensitive attributes* (e.g., race, gender, age) to preprocess the training data, regularize the model training or post-process the prediction results to derive fair predictions [33, 17, 27]. However, sensitive attributes are often hard to collect or properly protected due to people's increasing awareness of privacy and the legal compliance such as Electronic Communications Privacy Act (ECPA)[2] and General Data Protection Regulation (GDPR)[3] that restrict the direct access to sensitive attributes. Thus, analysts may only get access to

---

[1] www.forbes.com/sites/korihale/2021/09/02/ai-bias-caused-80-of-black-mortgage-applicants-to-be-denied
[2] www.bja.ojp.gov/program/it/privacy-civil-liberties/authorities/statutes/1285
[3] www.consumerfinance.gov/rules-policy/regulations/1002/5

2022 Trustworthy and Socially Responsible Machine Learning (TSRML 2022) co-located with NeurIPS 2022.

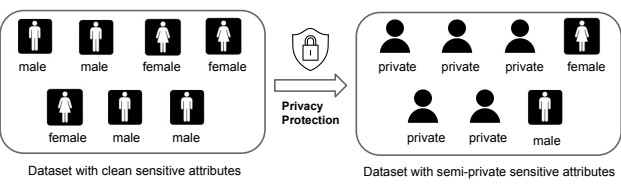

Figure 1: An illustration of the semi-private setting.

mostly *private* sensitive attributes processed by privacy mechanisms like Local Differential Privacy (LDP) [12], which is widely deployed by data analysts and can provide a strong privacy guarantee on the data by injecting noise [34, 19].

In practice, it is often possible to collect a small amount of sensitive attributes due to people's different privacy preferences (e.g., male users may be less sensitive about their age than female users) [24], organizations' desire to build effective models [7] or other certain incentives [38, 30]. Therefore, as shown in Figure 1, we propose to study a novel problem of *fair classification in a semi-private setting*, where most of the sensitive attributes (e.g., gender) are protected under certain privacy mechanisms and only very limited clean ones are available.

However, it is nontrivial to build fair machine learning models with mostly private sensitive attributes. First, private sensitive attributes are noisy, and directly applying conventional debiasing techniques on them can lead to sub-optimal performances. Some initial efforts have verified that noise-protected sensitive attributes may hurt the performance of conventional debiasing models [32, 42]. Second, it is unknown whether or not it is helpful to incorporate the limited clean sensitive attributes. Such limited instances with clean sensitive attributes are inadequate for training a fair classification model directly because the model can easily overfit the small amount of data [14]. It is under exploration to study the impact of combining the very limited data with clean sensitive attributes and the most private data. Third, the conventional model design cannot effectively leverage both the most private sensitive attributes and limited clean ones. Conventional debiasing models under the semi-private setting would treat each instance the same way. Since the clean sensitive attributes are limited, it is important to explore how to effectively exploit them.

Therefore, to address these challenges, we first conduct a preliminary study on the impact of privacy on fairness performance of debiasing and non-debiasing models. We found that the fairness performance of debiasing models is enhanced with a smaller noise rate on the sensitive attributes, which is determined by a constant *privacy budget*. This finding validates that the semi-private setting is beneficial for mitigating fairness bias. Based on the pilot study, we propose a novel framework FAIRSP for fair classification with semi-private sensitive attributes. It first *learns to correct* the noisy sensitive attributes under privacy guarantee by exploiting the limited clean ones. Specifically, our model learns a *correction matrix* for the noisy sensitive attributes by leveraging both the noisy ones and very limited clean ones. Then, it jointly models the corrected and clean data in an adversarial way for debiasing and prediction.

In summary, our main contributions are as follows:

- We empirically study the impact of privacy on fairness performance of debiasing and non-debiasing models.
- We study a novel and practical problem of fair classification with semi-private sensitive attributes.
- We provide a new end-to-end framework FAIRSP which simultaneously derives corrected sensitive attributes from private ones and learns a fair classifier with adversarial learning on both clean and corrected data.
- We conduct a theoretic analysis demonstrating that fairness can be achieved with mostly private sensitive attributes.
- We perform extensive experiments on real-world datasets to validate the effectiveness of the proposed *learning to correct* method for fair classification in the semi-private setting.

## 2   Assessing the Intersection of Privacy and Fairness

In this section, we first introduce the definition of local differential privacy. Then we conduct a preliminary study to assess the impact of privacy on fairness performances.

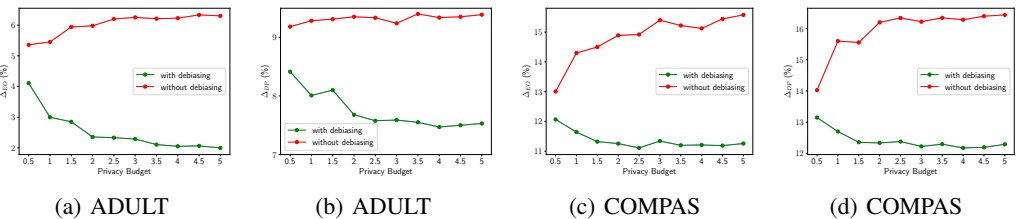

|  (a) ADULT | (b) ADULT | (c) COMPAS | (d) COMPAS |

Figure 2: Assessing the impact of privacy on fairness performances.

## 2.1 Local Differential Privacy Guarantee

The Local differential privacy (LDP) mechanism provides guarantees by directly injecting noises into data before aggregation. The formal Local differential privacy definition is as follows:

**Definition 1.** *Given $\epsilon > 0$, a randomized mechanism $\mathcal{M}$ satisfies $\epsilon$-local differential privacy, if for all possible pairs of users' private data $s_i$ and $s_j$, the following equation holds:*

$$\forall a \in \text{Range}(\mathcal{M}) : \frac{P(\mathcal{M}(s_i)=a)}{P(\mathcal{M}(s_j)=a)} \leq e^{\epsilon} \text{ where Range}(\mathcal{M}) \text{ denotes every possible output of } \mathcal{M}.$$

The parameter $\epsilon$ denotes the privacy budget to balance the utility and privacy guarantee of the model. A smaller $\epsilon$ represents stronger privacy guarantee and weaker utility. In order to ensure the privacy of sensitive attributes, we obtain the following lemma:

**Lemma 1.** *To achieve $\epsilon$-local differential privacy on the binary sensitive attribute, we can randomly flip the sensitive attributes with a probability of $p = \frac{1}{exp(\epsilon)+1}$.*

The detailed proof of the lemma refers to the lemma 3 in [32]. Based on the lemma, we can obtain *differentially private sensitive attributes* by flipping at a certain probability. If the flipping probability satisfies the condition that $p = \frac{1}{exp(\epsilon)+1}$, then the sensitive attributes have the $\epsilon$-*local differential privacy guarantee*.

## 2.2 The Impact of Privacy on Fairness

In this subsection, we investigate the impact of privacy on fairness. Here we adopt the notion of Local Differential Privacy (LDP) for the privacy guarantee. The original LDP mechanism focuses on the trade-off between privacy and utility. As for the impact of local differential privacy on fairness, we conduct two groups of preliminary experiments. For the first group of experiments, we study the impact of LDP on models like vanilla multi-layer per- ceptron (MLP) network without debiasing. For the second group of experiments, we study the impact of LDP on debiasing models such as adversarial debiasing. We conduct each group of experiments on two datasets ADULT and COMPAS, which are two typical fairness datasets. We set six privacy budgets for each group of experiments as 0.5, 1, 1.5, 2, 2.5, 3. We follow Lemma 1 to implement the LDP mechanism. From Figure 2, we have the following observations:

- **For vanilla models without debiasing, stronger privacy guarantee improves the fairness performance.** We can observe that with lower privacy budget, the vanilla models have better fairness performance on $\Delta_{EO}$ and $\Delta_{DP}$. With stronger privacy guarantee, the privacy budget decreases and the flipping rate increases, which means there is more noise injected into the sensitive attributes of the dataset. With more noise injected, the vanilla models cannot learn the explicit bias contained in the sensitive attributes.

- **For debiasing models, stronger privacy guarantee leads to worse fairness performance.** From Figure 2, we can see that with lower privacy budget, the debiasing models have a worse fairness performance on $\Delta_{EO}$ and $\Delta_{DP}$ on the two datasets. This is because the debiasing models need to explicitly leverage sensitive attributes for mitigating bias. With stronger privacy guarantee, there is a lower privacy budget and more noisy sensitive attributes, which causes that the debiasing models ineffective in mitigating the implicit bias contained in non-sensitive attributes.

Based on the preliminary experiments, **if we want to improve the fairness performance of debiasing models under privacy, one method is to reduce the noise in sensitive attributes**, which illustrates the benefit of the semi-private setting.

# 3 Fair Classification with Semi-Private Sensitive Attributes

In this section, we mainly introduce our method. More details on the setting and motivation of semi-private sensitive attributes are in the Appendix A.

# 4 Proposed Model - FAIRSP

Having defined the problem setting for fair classification in the presence of a small set of samples with the clean sensitive attribute and a large set of private ones, we now propose our approach to leverage them jointly to learn an end-to-end model. Next, we present the details of the proposed framework for fair classification with semi-private sensitive attributes.

## 4.1 Semi-Private Adversarial Debiasing

In our semi-private scenario, we have two distinct types of sensitive attributes: clean and private. Our objective is to build a framework that leverages the information from both clean and private samples and learn an underlying common representation that can induce the fair classification. Recently, adversarial debiasing has been proven to be effective in alleviating the bias of representations. In adversarial debiasing, an adversary is used to predict sensitive attributes from the representations of the classifier; the classifier is trained to learn representations that make the adversary unable to predict the sensitive attributes while keeping high accuracy in the classification. Directly applying adversarial debiasing is not feasible in our scenario since only limited clean sensitive attributes are available and most of the sensitive attributes are private (noisy). To this end, we propose to utilize a shared encoder layer to learn the embedding vector, which is fed into separate layers for debiasing.

Specifically, we first propose a label predictor to minimize the prediction error of the labels with the following objective function: $\min_{\theta_h, \theta_Y} \mathcal{L}_Y = \mathbb{E}_{(X,Y) \in \mathcal{D}} \ell(Y, f_{\theta_Y}(h(X)))$, where $f_{\theta_Y}$ is to predict the labels, $h(\cdot)$ is an embedding layer to encode the features into the latent representation space, and $\ell$ is a cross entropy loss. In addition, to learn fair representations and make fair predictions, we incorporate two adversaries $f_c$ and $f_p$ to predict the clean and private sensitive attributes, respectively. $h(\cdot)$ tries to learn the representation that can fool the adversaries. $f_{\theta_c}$ and $f_{\theta_p}$ are jointly optimized with the following objective function: $\min_{\theta_h} \max_{\theta_c, \theta_p} \mathcal{L}_a = \mathcal{L}_c + \alpha \mathcal{L}_p$, where $\alpha$ is a hyper-parameter that controls the relative importance of the loss functions computed over the data of clean and private sensitive attributes, and $\mathcal{L}_c$ and $\mathcal{L}_p$ are defined as follows:

$$\min_{\theta_h} \max_{\theta_c} \mathcal{L}_c = \mathbb{E}_{X \sim p(X|A_c=1)}[\log(f_{\theta_c}(h(X)))] + \mathbb{E}_{X \sim p(X|A_c=0)}[\log(1 - f_{\theta_c}(h(X)))] \quad (1)$$

$$\min_{\theta_h} \max_{\theta_p} \mathcal{L}_p = \mathbb{E}_{X \sim p(X|A_p=1)}[\log(f_{\theta_p}(h(X)))] + \mathbb{E}_{X \sim p(X|A_p=0)}[\log(1 - f_{\theta_p}(h(X)))] \quad (2)$$

where $\theta_c$ and $\theta_p$ are the parameters for the adversaries predicting the clean and private sensitive attributes. Finally, the overall objective function of adversarial debiasing for fair classification is a minmax function, where $\beta$ controls the importance of the sensitive attribute predictors: $\min_{\theta_h, \theta_Y} \max_{\theta_c, \theta_p} \mathcal{L}_{adv} = \mathcal{L}_Y - \beta(\mathcal{L}_c + \alpha \mathcal{L}_p)$.

## 4.2 Private Sensitive Attribute Correction

Since private sensitive attributes are noisy, directly applying adversarial debiasing on them may lead to sub-optimal results. An intuitive approach is to consider *learning to correct* these noisy sensitive attributes before feeding them into the above model (i.e., $\mathcal{L}_{adv}$). Inspired by the idea of learning a *Corruption Matrix* under the scenario with severe label noise and trusted data [23], we propose to leverage the limited clean sensitive attributes to learn a *Correction Matrix* to estimate the true sensitive attributes from the private ones without assuming the underlying noise distribution among the private sensitive attributes.

Specifically, given the dataset $\mathcal{D}_c = \{\mathcal{X}, \mathcal{A}_c, \mathcal{Y}\}$ with instances containing clean sensitive attributes of $l$ categories, and $\mathcal{D}_p = \{\mathcal{X}, \mathcal{A}_p, \mathcal{Y}\}$ with instances' sensitive attributes being private, we aim to estimate a sensitive attribute correction matrix $\mathbf{C} \in \mathbb{R}^{l \times l}$ to model the sensitive attribute correction process. We first train a sensitive attribute predictor $g$ on the private data $\mathcal{D}_p$ as follows: $g(X) = \hat{p}(A_p|X)$. Let $\mathcal{X}_m$ be the subset of $X$ with sensitive attribute $A_c = m$. Based on the Lemma 1, the

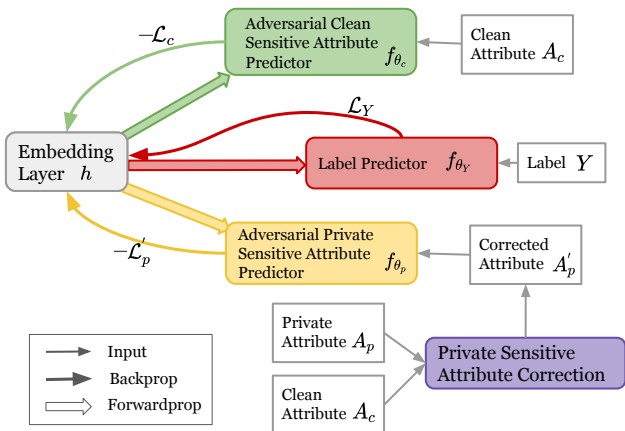

Figure 3: An illustration of the proposed framework FAIRSP. It consists of two major modules: (1) an semi-private adversarial debiasing module for learning fair classification; and (2) a private sensitive attribute correction module for correcting noisy sensitive attributes.

flipping probability in LDP is a constant determined by $\epsilon$. Thus, the noise distribution is independent from the sensitive attribute value. Then we can infer that $A_p$ is conditional independent from $A_c$ given $X$, i.e., $p(A_p|A_c, X) = p(A_p|X)$. Thus, we can estimate the correction matrix $\mathbf{C}$, where each element $\mathbf{C}_{mr}$ (the transition probability from $A_c = m$ to $A_p = r$) is calculated as follows:

$$\mathbf{C}_{mr} = \frac{1}{|\mathcal{X}_m|} \sum_{X \in \mathcal{X}_m} \hat{p}(A_p = r|X) = \frac{1}{|\mathcal{X}_m|} \sum_{X \in \mathcal{X}_m} \hat{p}(A_p = r|A_c = m, X) \approx p(A_p = r|A_c = m) \tag{3}$$

With the estimated $\mathbf{C}$, then we can train a new prediction model $g'(X) = \tilde{p}(A_c|X)$ solving the following optimization problem:

$$\min_{\theta_g, \theta_{g'}} \mathcal{L}_c = \mathbb{E}_{(X, A_c) \in \mathcal{D}_c} \ell(A_c, g'(x)) + \mathbb{E}_{(X, A_p) \in \mathcal{D}_p} \ell(A_p, \mathcal{C}^\mathrm{T} g'(X)) \tag{4}$$

where $\ell$ is a differentiable loss function to measure the prediction error, such as cross-entropy loss.

### 4.3  FAIRSP: Integrating Corrected Sensitive Attributes into Adversarial Debiasing

We aim to leverage the corrected sensitive attributes and integrate them to the adversarial debiasing algorithm for fair classification. Specifically, using the sensitive attribute correction network in Eqn. 4, we learn a sensitive attribute correction function $g'(X)$ that rectifies the private sensitive attributes for each instance $X \in \mathcal{D}_p$. We now obtain a set of instances with corrected sensitive attributes $\mathcal{D}'_p = \{X, A'_p, Y\}, \forall X \in \mathcal{D}_p$ and $A'_p = g'(X)$. Note that the sensitive attribute correction network reduces the noise but the rectified sensitive attributes could still be erroneous. Essentially, we first feed the instances with private sensitive attributes $\mathcal{D}_p$ into the sensitive attributes correction network to obtain the rectified instances $\mathcal{D}'_p$. These rectified instances are used as the input for the adversarial debiasing of private sensitive attributes (i.e., Eqn. 2). Formally, the objective function of the adversary $f_{\theta_p}$ is as defined as follows:

$$\min_{\theta_h} \max_{\theta_p} \mathcal{L}'_p = \mathbb{E}_{X \sim p(X|A'_p = 1)}[\log(f_{\theta_p}(h(X)))] + \mathbb{E}_{X \sim p(X|A'_p = 0)}[\log(1 - f_{\theta_p}(h(X)))] \tag{5}$$

The overall objective function of our final model FAIRSP is:

$$\min_{\theta_h, \theta_Y} \max_{\theta_c, \theta_p} \mathcal{L} = \mathcal{L}_Y - \beta(\mathcal{L}_c + \alpha \mathcal{L}'_p) \tag{6}$$

## 5  Experiments

In this section, we conduct experiments to evaluate the performance of our method. More details on the experiment analysis are in the Appendix C.

Table 1: The performance comparison for fair classification under semi-private setting.

| Datasets | Metric | Vanilla | RemoveS | RNF [15] | FariRF [47] | Clean | Private | C+P | FAIRSP |
|---|---|---|---|---|---|---|---|---|---|
| **ADULT** | Acc.(%) | 84.8±0.2 | 84.9±0.3 | 83.5±1.2 | 84.0±0.5 | 84.9±0.4 | 84.7±0.3 | 84.8±0.5 | 84.7±0.4 |
| | F1(%) | 65.4±0.7 | 64.8±0.8 | 63.3±0.8 | 63.5±0.7 | 64.6±0.7 | 64.6±0.3 | 64.8±0.6 | 64.5±0.7 |
| | $\Delta_{DP}$(%) | 9.1±0.4 | 8.4±0.2 | 8.3±1.0 | 8.2±0.3 | 8.4±0.4 | 8.4±0.3 | 8.1±0.2 | **7.8±0.3** |
| | $\Delta_{EO}$(%) | 5.3±1.0 | 4.1±1.1 | 4.0±0.5 | 3.5±0.8 | 4.1±1.0 | 4.1±1.2 | 3.4±1.4 | **2.3±1.2** |
| **COMPAS** | Acc.(%) | 67.0±0.6 | 67.3±0.8 | 66.9±0.8 | 66.3±0.7 | 67.2±0.6 | 67.1±0.7 | 67.2±0.6 | 67.0±0.6 |
| | F1(%) | 64.3±0.9 | 64.2±1.2 | 63.5±0.9 | 63.2±0.5 | 64.8±1.0 | 64.6±1.1 | 63.9±1.1 | 63.8±1.4 |
| | $\Delta_{DP}$(%) | 13.8±1.1 | 13.0±0.4 | 13.1±0.6 | 13.8±2.4 | 13.1±0.5 | 13.0±0.4 | 12.9±0.2 | **12.7±0.5** |
| | $\Delta_{EO}$(%) | 12.8±1.4 | 12.2±0.6 | 12.3±1.3 | 15.3±1.2 | 12.3±0.8 | 12.1±0.7 | 12.2±0.5 | **12.1±0.6** |
| **MEPS** | Acc.(%) | 86.1±0.1 | 86.1±0.2 | 85.8±0.1 | 85.9±0.2 | 86.1±0.1 | 86.0±0.1 | 86.1±0.1 | 86.0±0.1 |
| | F1(%) | 48.5±2.0 | 49.9±1.6 | 49.5±1.5 | 47.0±1.9 | 50.6±1.6 | 50.8±2.3 | 48.8±1.8 | 47.3±1.7 |
| | $\Delta_{DP}$(%) | 4.5±0.5 | 4.7±0.5 | 4.8±0.3 | 4.9±1.0 | 4.8±0.6 | 4.8±0.7 | 4.4±0.4 | **4.1±0.8** |
| | $\Delta_{EO}$(%) | 4.5±1.0 | 4.6±1.1 | 4.8±0.9 | 4.7±1.3 | 4.4±1.2 | 4.5±1.1 | 4.3±0.7 | **4.0±1.2** |

We compare FAIRSP with various baselines on three benchmark datasets for fair classification. For each experiment, we select five random seeds to partition the original dataset into a training set and a test set. We randomly select 80% of the training set as the private training set and the others as the non-private part (i.e., clean ratio =20%). For the private training set, we set the privacy budget $\epsilon$ as 0.5, which means the flipping probability on sensitive attributes is around 38% according to the Lemma 1. The average performance and standard deviation over five times are reported in Table 1. We can make following observations:

- In general, we observe that FAIRSP can better improve fairness performance of $\Delta_{DP}$ and $\Delta_{EO}$ without causing significant drop in prediction performance on three datasets, compared to other baselines. For example, compared with recently proposed state-of-the-art debiasing model RNF, FAIRSP has achieved 42.5% improvement in terms of $\Delta_{EO}$ and 6.0% on $\Delta_{DP}$ on ADULT.

- We observe that conventional debiasing methods that directly leverage sensitive attributes are generally ineffective under the semi-private scenario. For example, comparing RNF with RemoveS model, we can observe that their fairness performances are similar w.r.t. $\Delta_{DP}$ and $\Delta_{EO}$ on three datasets.

- The proposed private sensitive attribute correction can help better utilize the limited clean sensitive attributes for improving fairness. For example, FAIRSP has 32.3% and 3.7% improvement over "Clean+Private" on ADULT by $\Delta_{DP}$ and $\Delta_{EO}$ respectively. The reason is that "Clean+Private" directly utilizes the private sensitive attributes; while the corrected sensitive attributes contain less severe noise and can better guide the debiasing process.

- We can see that it is important to leverage sensitive attributes to ensure fairness and maintain prediction performance in the semi-private scenario. For example, FAIRSP has significant improvement w.r.t. Accuracy, F1, $\Delta_{DP}$ and $\Delta_{EO}$ than FairRF consistently, which does not leverage the sensitive attributes directly.

- Exploiting the limited clean sensitive attributes and private ones jointly is important for debiasing in the semi-private setting. We can generally observe that FAIRSP > Clean+Private > Clean ≈ Private ≈ RemoveS > Vanilla for debiasing performances. First, FAIRSP and "Clean+Private" perform better than the other three baselines shows that leveraging both clean and private sensitive attributes is necessary. Second, the observation that "Clean", "Private" and RemoveS perform similar indicates that only relying on the clean data or noisy data is less effective for debiasing.

## 6   Conclusion and Future Work

In this paper, we study a novel problem of fair classification with semi-private sensitive attributes. We develop an end-to-end adversarial debiasing model FAIRSP to jointly learn from a small amount of instances with clean sensitive attributes and a large amount of instances with private ones. We provide a theoretic analysis to demonstrate that we can learn a fair model prediction under mild assumptions on privacy. Extensive experimental results on real-world datasets demonstrate the effectiveness of the proposed framework to achieve better fair classification performance compared to existing approaches. For future work, first, we can study a more general semi-private fairness setting in a variety of data types such as text and graphs, and analyze the fairness and privacy guarantee on these data. Second, we will investigate the fairness under privacy in distributed machine learning algorithms such as federated learning and split learning.

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

# A  More Details on Method

## A.1  Semi-Private Sensitive Attributes

Training fair classification models with only private sensitive attributes is challenging since most conventional debiasing methods rely on the sensitive information for mitigating fairness bias. The preliminary experiments have demonstrated that privacy noise on sensitive attributes can severely hurt the fairness performance. Therefore, we study a more practical problem where most of the sensitive attributes are private and only very limited ones are clean because it is often possible to obtain a small amount of clean sensitive attributes as discussed earlier [24, 7, 38, 30]. To better reflect the reality in our experimental setup, we keep at most 20% of all the samples to be non-private (see Section C.1.1 for more details). Such non-private samples provide the clean sensitive attributes and will not overlap with those private ones.

## A.2  Problem Statement

We first introduce the notations of this paper and then give the formal problem definition. Let $\mathcal{D} = \{\mathcal{X}, \mathcal{A}, \mathcal{Y}\}$ denote the data, where $\mathcal{X}$, $\mathcal{A}$, and $\mathcal{Y}$ represent the set of data samples, sensitive attributes, and corresponding labels. For the sensitive attributes $\mathcal{A}$, it consists of a small number of clean ones $\mathcal{A}_c$, and a large number of private ones $\mathcal{A}_p$, i.e., $\mathcal{A} = \mathcal{A}_c \cup \mathcal{A}_p$. Usually, the size of clean sensitive attributes is much smaller than the size of private sensitive attributes due to the increasing awareness of privacy of users and service providers. The private sensitive attributes $\mathcal{A}_p$ can be collected from user-obfuscated input or a third-party privacy-preserving algorithm. In this paper we adopt LDP mechanism [12].

Following the existing works of fair classification [5, 35], we evaluate the performance of fairness using metrics such as Equal Opportunity and Demographic Parity. Without loss of generality, we consider the binary classification. Equal Opportunity requires that the probability of positive instances with arbitrary sensitive attributes $A$ being assigned to a positive outcome are equal: $\mathbb{E}(\hat{Y} \mid A = a, Y = 1) = \mathbb{E}(\hat{Y} \mid A = b, Y = 1)$, where $\hat{Y}$ is predicted label. Demographic Parity requires the behavior of prediction model to be fair on different demographic groups. Concretely, it requires that the positive rate across sensitive attributes are equal: $\mathbb{E}(\hat{Y} \mid A = a) = \mathbb{E}(\hat{Y} \mid A = b), \forall a, b$. The problem of fair classification with semi-private sensitive attributes is formally defined as follows:

> **Problem Statement:** Given the training data $\mathcal{D}$ with a limited number of clean sensitive attributes and a large amount of private sensitive attributes, learn an effective classifier that generalizes well to unseen instances, while satisfying the fairness criteria such as demographic parity.

## A.3  The Optimization of FAIRSP

---

**Algorithm 1** Training process of FAIRSP

---

**Require:** Clean data $\mathcal{D}_c = \{X, A_c, Y\}$, private data $\mathcal{D}_p = \{X, A_p, Y\}$, $\alpha$, $\beta$
**Ensure:** Learn a fair classifier $f_{\theta_Y}$
1: // Correcting private sensitive attributes
2: Learn a classifier $g$ on $\mathcal{D}_p$
3: Initialize correction matrix $\mathbf{C} \in \mathbf{R}^{l \times l}$
4: Estimate the correction matrix as in Eqn.3
5: Initialize a new classifier $g'$
6: Train $\ell(A_c, g'(X))$ on $\mathcal{D}_c$, $\ell(A_p, \mathcal{C}^{\mathrm{T}} g'(X))$ on $\tilde{\mathcal{D}}$ as in Eqn. 4
7: Generate data with corrected sensitive attributes $\mathcal{D}'_P = \{X, g'(X), Y\}$
8: // Incorporate corrected sensitive attributes for FAIRSP training
9: Conduct adversarial training with loss function as Eqn. 6 on $\mathcal{D}'_P$ and $\mathcal{D}_c$.
10: Return classifier $f_{\theta_Y}$ for fair prediction

---

We adopt the mini-batch gradient descent with Adadelta [45] optimizer to learn the parameters. As shown in Algorithm 1, we first train the sensitive attribute correction network to obtain the data with

corrected sensitive attributes $\mathcal{D}'_p$. To this end, we train a predictor $g$ on the data with private sensitive attributes $\mathcal{D}_p$ and estimate the sensitive attribute correction matrix $\mathbf{C}$ as in Eqn. 3. Thereafter, we train a new predictor $g'$ with the correction matrix $\mathbf{C}$ on the private data, and obtain the data with corrected sensitive attributes $\mathcal{D}'_p$. Finally, we train the adversarial debiasing model for classification in both $\mathcal{D}'_P$ and $\mathcal{D}_c$.

## B  A Theoretic Analysis on Fairness Guarantee

In this section, we perform a theoretic analysis of the fairness guarantee under the proposed FAIRSP. The model essentially integrates two major modules: (1) the private sensitive attribute correction ($\mathcal{L}_c$ in Eqn. 4); and (2) the semi-private adversarial debiasing with the corrected sensitive attributes ($\mathcal{L}_c + \alpha \mathcal{L}'_p$ in Eqn. 6).

First, to understand the theoretic guarantee for the semi-private adversarial debiasing, we need to consider the induced noise in the large amount of private sensitive attributes as they are non-negligible. We can obtain the condition of the global optimum of the adversaries by Theorem 1.

**Theorem 1.** *The global optimum of the adversaries in $\mathcal{L}$ ($f_{\theta_c}$ and $f_{\theta_p}$) can be achieved if and only if $p(X|A_c = 1) = p(X|A_c = 0)$ and $p(X|A'_p = 1) = p(X|A'_p = 0)$.*

*Proof.* This is because according to the Proposition 1. in [20], the adversaries can reach to the optimal $f^*_{\theta_c} = \frac{p(X|A_c=1)}{p(X|A_c=1)+p(X|A_c=0)}$ and $f^*_{\theta_p} = \frac{p(X|A'_p=1)}{p(X|A_c=1)+p(X|A'_p=0)}$. Then the optimal of the min-max game can be written as follows:

$$R = \mathbb{E}_{X \sim p(X|A_c=1)}[\log \frac{p(X|A_c=1)}{p(X|A_c=1)+p(X|A_c=0)}] \tag{7}$$
$$+ \mathbb{E}_{X \sim p(X|A_c=0)}[\log \frac{p(X|A_c=0)}{p(X|A_c=1)+p(X|A_c=0)}]$$
$$+ \alpha \mathbb{E}_{X \sim p(X|A'_p=1)}[\log \frac{p(X|A'_p=1)}{p(X|A'_p=1)+p(X|A'_p=0)}]$$
$$+ \alpha \mathbb{E}_{X \sim p(X|A'_p=0)}[\log \frac{p(X|A'_p=0)}{p(X|A'_p=1)+p(X|A'_p=0)}]$$
$$= -(1+\alpha)\log(4) + 2 \cdot JSD(p(X|A_c=1)||X|A_c=0)$$
$$+ 2\alpha \cdot JSD(p(X|A'_p=1)||X|A'_p=0) \tag{8}$$

Since $\alpha$ is a non-negative hyper-parameter and the Jensen–Shannon divergence (JSD) between two distributions is always non-negative and zero only when they are equal, so if and only if $p(X|A'_p = 1) = p(X|A'_p = 0)$ and $p(X|A_c = 1) = p(X|A_c = 0)$, the above equation will reach to the minimum. $\square$

Next, we theoretically show in Theorem 2 that under mild conditions, we can satisfy fairness metrics such as demographic parity if reaching the global optimum.

**Theorem 2.** *Let $\hat{Y}$ denotes the predicted labels, if*
*(1) For all $X \in \mathcal{D}'_p$, the corrected sensitive attributes $A'_p$ and $X$ are independent conditioned on their ground-truth sensitive attributes $\tilde{A}_p$, i.e., $p(A'_p, X|\tilde{A}_p) = p(A'_p|\tilde{A}_p)p(X|\tilde{A}_p)$;*
*(2) The corrected sensitive attributes are not random, i.e., $p(A'_p = 1|\tilde{A}_p = 1) \neq p(A'_p = 0|\tilde{A}_p = 0)$)*
*If $\mathcal{L}$ reaches the global optimum, the label prediction $f_{\theta_Y}$ will achieve demographic parity, i.e., $p(\hat{Y}|\tilde{A}_p = 0) = p(\hat{Y}|\tilde{A}_p = 1)$ and $p(\hat{Y}|A_c = 0) = p(\hat{Y}|A_c = 1)$*

*Proof.* We first explain the two assumptions: (1) since we use the objective function in $\mathcal{L}_c$ to derive the corrected sensitive attributes $A'_p$, and learn the latent presentation through $h(\cdot)$, between which they do not share any parameters; it generally holds that $A'_p$ is independent with the representation of $X$, i.e., $p(A'_p = 1|\tilde{A}_p = 1) \neq p(A'_p = 0|\tilde{A}_p = 0)$); (2) Since we are using adversarial learning

to learn an effective estimator $f_{\theta_p}$ for sensitive attributes, it is reasonable to assume that it does not produce random prediction results. We then prove Theorem 2 as follows: since $p(A_p^{'} = 1|\tilde{A}_p = 1) \neq p(A_p^{'} = 0|\tilde{A}_p = 0))$, we have $p(X|A_p^{'}, \tilde{A}_p) = p(X|\tilde{A}_p)$. When the algorithm converges, we have $p(X|A_p^{'} = 1) = p(X|A_p^{'} = 0)$, which is equivalent with $\sum_{\tilde{A}_p} p(X, \tilde{A}_p|A_p^{'} = 1) = \sum_{\tilde{A}_p} p(X, \tilde{A}_p|A_p^{'} = 0)$. Therefore,

$$\sum_{\tilde{A}_p} p(X|\tilde{A}_p)p(X, \tilde{A}_p|A_p^{'} = 1) = \sum_{\tilde{A}_p} p(X|\tilde{A}_p)p(X, \tilde{A}_p|A_p^{'} = 0)$$

Based on the above equation, we can get,

$$\frac{p(X|\tilde{A}_p = 1)}{p(X|\tilde{A}_p = 0)} = \frac{p(\tilde{A}_p = 0|A_p^{'} = 1) - p(\tilde{A}_p = 0|A_p^{'} = 0)}{p(\tilde{A}_p = 1|A_p^{'} = 0) - p(\tilde{A}_p = 1|A_p^{'} = 1)} = 1 \tag{9}$$
$$\Rightarrow p(X|\tilde{A}_p = 1) = p(X|\tilde{A}_p = 0)$$

Since we already proof that $p(X|A_c = 1) = p(X|A_c = 0)$, and $\hat{Y} = f_{\theta_Y}(X)$, we can get $p(\hat{Y}|\tilde{A}_p = 1) = p(\hat{Y}|\tilde{A}_p = 0)$ and $p(\hat{Y}|A_c = 1) = p(\hat{Y}|A_c = 0)$, which is the demographic parity. $\qquad\square$

## C   More on Experiment Analysis

In this section, we conduct experiments to evaluate the performance of our method. In these experiments, we try to answer the following research questions:

- **RQ1**: Can FAIRSP obtain fair predictions with mostly private sensitive attributes?
- **RQ2**: How does the private sensitive attribute correction affect the performance of prediction and fairness?
- **RQ3**: What is the impact of the amount of data with clean sensitive attributes?
- **RQ4**: How does the degree of privacy budget impact the fair classification performance?

Note that **RQ1** has been answered in Section 5.

### C.1   Experimental Settings

#### C.1.1   Datasets

We conduct experiments on three typical datasets for fair classification: COMPAS [26], ADULT [4] and MEPS [11].

- **COMPAS**[4]: This dataset describes the task of predicting the recidivism of individuals in the United States.
- **ADULT**[5]: This dataset contains personal yearly records. The task is to predict whether an individual's income exceeds 50k.
- **MEPS**[6]: This dataset contains records of The Medical Expenditure Panel Survey (MEPS), which is a set of large-scale medical surveys across the United States. The task is to predict whether or not a person would have a "high" utilization.

We report results on the test set and all experiments are repeated for 5 times and the average result and standard deviation are reported in Table 1. For each run, the dataset is randomly divided into train set and test set. The set of random seeds for five runs is {5, 7, 11, 19, 29}. We compare different methods and techniques at different values of the *clean ratio* defined as:

$$\text{clean ratio} = \frac{\#\text{samples w/ clean SA}}{\#\text{samples w/ clean SA} + \#\text{samples w/ private SA}} \tag{10}$$

---

[4]https://github.com/propublica/compas-analysis
[5]https://archive.ics.uci.edu/ml/machine-learning-databases/adult/
[6]https://meps.ahrq.gov/mepsweb/

| Data | SA | # Train | | # Test |
|---|---|---|---|---|
| | | Clean (20%) | Private (80%) | |
| **ADULT** | Gender | 4,884 | 19,536 | 24,421 |
| **COMPAS** | Race | 611 | 2,446 | 3,058 |
| **MEPS** | Race | 1,573 | 6,292 | 7,866 |

Table 2: The statistics of the datasets. Clean refers to true sensitive attributes, whereas Private refers to the private ones with LDP. SA refers to sensitive attributes.

### C.1.2 Baselines

We compare the proposed FAIRSP with four types of baselines. The first type of baseline is a vanilla multi-layer perceptron (MLP) network (1,2). The second type is a state-of-the-art debiasing model that explicitly utilizes the sensitive information (3). The third type is a debiasing model that do not need the sensitive information (4). For the fourth type of baselines, we implement a adversarial debiasing model with three kinds of training strategies including "Clean", "Private" and "Clean+Private" (5,6,7), which are utilized to illustrate the effectiveness of our proposed private sensitive attribute correction method: (1) Vanilla: A classifier like MLP without any debiasing method.; (2) RemoveS: Directly removing the sensitive attributes in the input; (3) RNF [15]: A recently proposed state-of-the-art debiasing method utilizing mixup for debiasing; (4) FairRF [47]: A debiasing method that does not need any sensitive attributes, which can also be applied in our proposed semi-private scenario; (5) Clean: An adversarial debiasing model on only the instances with clean sensitive attributes; (6) Private: An adversarial debiasing model on only the instances with private sensitive attributes; (7) C+P: We simply merge both the sets (essentially treating the private sensitive attributes as the clean ones) and use them together for training an adversarial debiasing model.

### C.1.3 Evaluation Metrics

Following existing work on fair classification, we measure the classification performance with Accuracy (Acc.) and F1, and the fairness performance based on *Demographic Parity* and *Equal Opportunity* [35].

- Demographic Parity: it requires each demographic group has the same chance for a positive outcome: $\mathbb{E}(\hat{Y}|A=1) = \mathbb{E}(\hat{Y}|A=0)$. We report the difference of each group's demographic parity:

$$\Delta_{DP} = |\mathbb{E}(\hat{Y}|A=1) - \mathbb{E}(\hat{Y}|A=0)| \tag{11}$$

- Equal Opportunity: it requires the true positive rate of different groups is equal: $\mathbb{E}(\hat{Y}|A=1, Y=1) = \mathbb{E}(\hat{Y}|A=0, Y=1)$. We report the difference of each sensitive group's equal opportunity:

$$\Delta_{EO} = |\mathbb{E}(\hat{Y}|A=1, Y=1) - \mathbb{E}(\hat{Y}|A=0, Y=1)| \tag{12}$$

Note that demographic parity and equal opportunity measure the fairness performance in different ways. The fairness performance is better with smaller values of $\Delta_{EO}$ and $\Delta_{DP}$.

### C.2 Assessing the Impact of Private Sensitive Attribute Correction

Now we investigate the impact of private sensitive attribute correction, to answer **RQ2**. We keep the setting of the training set and test set division the same as the main results in Table 1, and show the results on ADULT dataset as we have similar observations on COMPAS. We set the privacy budget $\epsilon$ as 0.5 and 1 for the training set to derive private sensitive attributes, and the corresponding noisy ratio on sensitive attributes are 38% and 27% accordingly. The average performance and standard deviation for five rounds are reported in Table 3. We can make following observations:

- The proposed correction strategy on private sensitive attributes is effective in improving debiasing performance consistently regardless of the privacy budget. For example, when the privacy budget $\epsilon$ is 0.5, FAIRSP has achieved a 23.3% and 6.0% improvement in terms of $\Delta_{EO}$ and $\Delta_{DP}$, comparing to FAIRSP without private attributes correction.

| Privacy Budget | Metric | FAIRSP w/o Corr | FAIRSP |
|---|---|---|---|
| $\epsilon = 0.5$ | Acc. (%) | 84.8±0.2 | 84.7±0.4 |
| | F1 (%) | 64.7±1.2 | 64.5±0.7 |
| | $\Delta_{DP}$ (%) | 8.3±0.8 | **7.8±0.3** |
| | $\Delta_{EO}$ (%) | 3.0±1.2 | **2.3±1.2** |
| $\epsilon = 1$ | Acc. (%) | 84.7±0.4 | 84.6±0.4 |
| | F1 (%) | 64.4±0.5 | 64.1±0.7 |
| | $\Delta_{DP}$ (%) | 7.9±0.4 | **7.5±0.1** |
| | $\Delta_{EO}$ (%) | 2.1±2.0 | **1.3±1.4** |

Table 3: The impact of private sensitive attribute correction.

- The proposed private sensitive attribute correction method does not cause a significant drop in prediction performance. For example, when privacy budget $\epsilon$ is 1, FAIRSP has achieved comparable performance with FAIRSP without correction in terms of both Accuracy and F1 metrics.

## C.3 Impact of Private Data Ratio

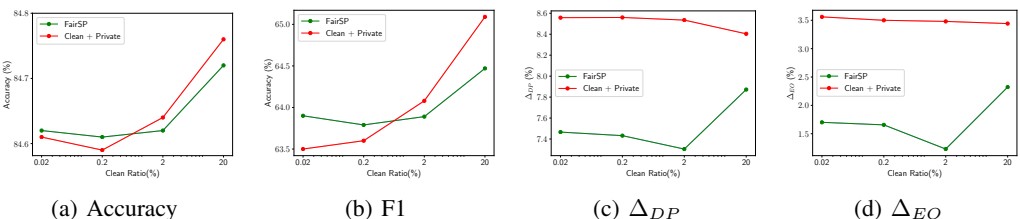

(a) Accuracy      (b) F1      (c) $\Delta_{DP}$      (d) $\Delta_{EO}$

Figure 4: The impact of clean data ratio on prediction and debiasing performances on ADULT.

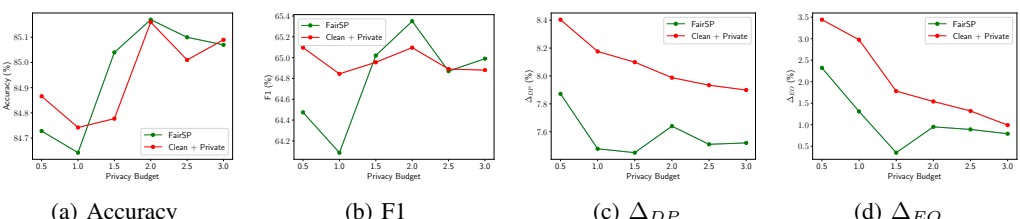

(a) Accuracy      (b) F1      (c) $\Delta_{DP}$      (d) $\Delta_{EO}$

Figure 5: The impact of privacy budget $\epsilon$ on prediction and debiasing performances on ADULT.

In this subsection, we investigate the impact of different private data ratios, to answer **RQ3**. We show the experimental results on ADULT dataset and we observe similar trends on COMPAS dataset. We conduct four groups of experiments with different clean data ratios (defined as in Eqn. 10) as in the range of [0.02%, 0.2%, 2%, 20%]. Each group of experiments have the same privacy budget $\epsilon$ as 0.5. For each clean data ratio, we compare our proposed model FAIRSP with "Clean+Private" and demonstrate the prediction metric Accuracy, F1 as well as fairness metric $\Delta_{DP}$ and $\Delta_{EO}$. For each experiment, we run five times and report the average result in Figure 4. From the figure, we can make following observations:

- In general, we can observe from Figure 4 (c) (d) that our proposed FAIRSP has a consistent improvement in fairness performance by a large margin compared with the baseline "Clean+Private" consistently at different clean data ratios.

- It should be noted that our model FAIRSP performs stable on fairness when the clean data ratios are smaller. The performance of the baseline "Clean+Private" on fairness slightly drops with smaller clean data ratio, which is expected because smaller clean data ratio means the dataset contains less useful information about sensitive attributes. However, our proposed model FAIRSP has stable fairness performance on both $\Delta_{DP}$ and $\Delta_{EO}$ with a smaller clean data ratio.

- Even with extreme small clean data ratio such as 0.02%, FAIRSP maintains relatively stable fairness performances. This indicates that the sensitive attributes correction is still effective with very limited clean data and under strong privacy requirements.

- From Figure 4 (a) (b), we observe that the proposed FAIRSP demonstrates comparable prediction performance with the baseline "Clean+Private" regardless of different clean data ratios. The prediction performance of both models drops slightly with smaller clean data ratio. This is because with smaller clean data ratio, there is a larger portion of data with private sensitive attributes.

## C.4 Impact of Privacy Budget

To answer **RQ4**, in this subsection, we investigate the impact of different privacy budgets on the fair classification performances. We report the experimental results on ADULT dataset and similar trends are observed on COMPAS. We conduct six groups of experiments with different privacy budget $\epsilon$ as {0.5, 1.0, 1.5, 2.0, 2.5, 3.0}, which means the flipping probability on sensitive attributes is around {38%, 27%, 18%, 12%, 7%, 4%} accordingly. The clean data ratio for each group of experiment is 20%. For each Privacy Budget $\epsilon$, we compare our proposed model FAIRSP and baseline "Clean+Private" with performance metrics Accuracy, F1 and fairness metrics $\Delta_{DP}$ and $\Delta_{EO}$. From Figure 5, we have the following observations:

- In general, we can observe from Figure 5 (c) (d) that our proposed model FAIRSP performs consistently better compared to "Clean+Private". With a larger privacy budget $\epsilon$, the fairness performances of both models are improved. This is because the dataset has fewer noisy sensitive attributes under larger $\epsilon$.
- With a larger privacy budget $\epsilon$, we can observe that the gap between FAIRSP and "Clean+Private" is narrower. This may be due to the estimation accuracy of the private sensitive attributes in FAIRSP becomes less effective when the flipping probability on sensitive attributes is small enough, which also indicates that FAIRSP is more effective when the privacy guarantee is strong.
- From Figure 5 (a) and (b), we see that FAIRSP has comparable classification performances on Accuracy and F1 with the baseline "Clean+Private" regardless of privacy budget $\epsilon$. With a larger privacy budget $\epsilon$, the performances of the both models increase because a larger privacy budget $\epsilon$ indicates fewer noisy sensitive attributes in the training data.

## C.5 Parameter Sensitivity Analysis

We now explore the parameter sensitivity of the two important hyperparameters of our model: $\alpha$ controls the impact of the adversarial private sensitive attribute predictor, while $\beta$ controls the influence of the adversary to the debiasing. We vary $\alpha$ in [0.6, 0.7, 0.8, 0.9, 1.0] and $\beta$ from [0.6, 0.7, 0.8, 0.9, 1.0]. The results are shown in Fig. 6. From the figure, we can observe that: (1) The performances of Accuracy and F1 are relatively consistent in the range, and the performance trend of $\Delta_{DP}$ and $\Delta_{EO}$ are also similar. (2) When $\beta$ is larger, the prediction performance drops and fairness performance is improved, which illustrates the trade-off between fairness and prediction performances. (3) Based on the experiments, we can achieve optimal fairness performance and comparable accuracy performance when selecting both $\alpha$ and $\beta$ as 1.0.

# D    Related Work

In this section, we briefly describe the related work on (1) Fairness in machine learning; and (2) Differential privacy in machine learning.

**Fairness in Machine Learning** Recent research on fairness in machine learning has drawn significant attention to develop effective algorithms to achieve fairness and maintain good prediction performance. Existing methods generally focus on individual fairness [28, 10, 8] or group fairness [22, 46]. Other niche notions of fairness include subgroup fairness [29] and Max-Min fairness[31]. The majority of existing debiasing techniques have been applied at different stages of a machine learning model [36] including *pre-processing* [27], *in-processing* [2, 6] and *post-processing* approaches [17]. Such machine learning methods generally require the access to sensitive attributes, which is often infeasible in practice. Very few recent work study fairness with limited or private sensitive attributes available. For example, Zhao *et al.* explore and utilize the related features as proxies of the sensitive attributes to improve fairness performances when sensitive attributes are unknown [47]. Dai *et al.* propose to achieve fairness on graph neural networks when the sensitive attributes are limited and private [13]. However, these methods may design for a specific type of data (e.g., graphs) or not considering the sensitive attributes being private.

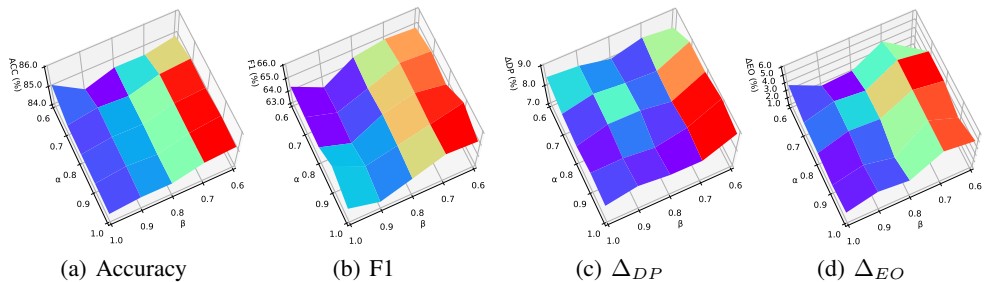

Figure 6: The parameter sensitive analysis of FAIRSP.

**Differential Privacy in Machine Learning.** Differential privacy (DP) [16, 18] is a widely adopted approach to provide strong privacy guarantee regardless of the adversaries' prior knowledge [34], which can protect user privacy in various machine learning tasks including supervised learning and unsupervised learning [25, 19, 1, 41]. Recently, local differential privacy (LDP) has been extensively studied in the distributed setting such that private data can be locally perturbed without a trusted aggregator [12, 43]. Due to the inherent connection of privacy and fairness (e.g., protecting or debiasing on user sensitive attributes), several recent work look into trade-offs and mutual risks between privacy and fairness [3, 9]. Other approaches aim to ensure both DP and fairness while preserving good utility [44], or learn fair models with only private data [37, 32, 42]. However, these initial efforts assume that only noisy sensitive attributes are available and there is a specific type of noise model between the true sensitive attributes and noisy ones.

