# OpenReview forum: "When Fairness Meets Privacy: Fair Classification with Semi-Private Sensitive Attributes"
_NeurIPS.cc/2022/Workshop/TSRML — TSRML2022_

### Official Review · Reviewer_Krji · 2022-10-19

**Overall Rating:** 6

**Summary:**

The paper studies how to make fair predictions in a semi-private setting, where most of the sensitive attributes are private (with attribute noise) and a small set of attributes are clean. For the sensitive attributes part, the paper uses a correction matrix to correct the noisy attributes. For the learning part, the paper uses an adversarial training method to ensure fairness.

**Strengths:**

The paper uses the correction method from learning with noisy labels to reduce the negative effect of attribute noise caused by differential privacy, and uses the adversarial learning method to ensure fair training.

**Weaknesses:**

1. The correction part should be compared with the method in [32] since [32] also used a correction matrix.
2. It is interesting to show how accurate the correction matrix is estimated.
3. Is the current estimation of correction matrix relies on the clean attributes? If so, what if the private part is not iid as the clean part?
4. Minor question: if we only want to guarantee differential privacy, can we just assume the attribute noise generation matrix is known?

**Overall Recommendation:**

The paper is relevant to the topic of this workshop although the proposed method may be similar to existing works.

**Review Confidence:**

5: The reviewer is absolutely certain that the evaluation is correct and very familiar with the relevant literature

---

### Official Review · Reviewer_AuCu · 2022-10-19
**Impact topic with a novel idea**

**Overall Rating:** 7

**Summary:**

The paper studied the fairness performance in semi-private settings. The training dataset can be split into data with clean and private sensitive attributes, where the private sensitive attributes are noisy. This is called a semi-private setting. To leverage the requirements of both privacy and fairness, they proposed a framework FAIRSP that trains private sensitive attributes predictor, clean sensitive attributes predictor, and class label predictor jointly. Extensive experiments were conducted on three datasets and improved fairness performance. They also did a theoretical analysis of fairness guarantees under semi-private settings.

**Strengths:**

The idea is novel: fairness performance under semi-private settings.

The motivation is clearly illustrated: the experiment in figure 2  gives us an example to introduce the motivation.

The storyline is well-crafted: since we provide a privacy guarantee via injecting noise, then we could denoise the sensitive attributes.

**Weaknesses:**

The plots are not informative. For example, in figure 2, instead of with debiasing and without debiasing, the authors could have the vanilla model and debiasing model. Also, the authors may consider putting more plots in the main body for experiments.

Some definitions are missing, making formulas inconsistent. For example, in the formula (3), $p$ and $p_hat$ are not defined. Someone may confuse what is the difference between them. Variables r is not defined. Also, formulas (3) and (4) are the multi-classification case, and (5) and (6) become the binary-classification case. For example, $A_p=r$ in (3), and ${A’}_p=1$ in (5). This is not consistent at all, and readers may get confused without proper illustration.

Finally, in the paper, you assume the private attributes contain noise. However, what if the private attributes are completely missing or unknown? In this case, would this FAIRSP framework still work?

**Overall Recommendation:**

The idea is novel, and the storyline is well-crafted. Combined two ideas of privacy and fairness together. There are some minor issues that need to be corrected in the formulas, but this work is still worth noticing.

**Review Confidence:**

4: The reviewer is confident but not absolutely certain that the evaluation is correct

---

### Official Review · Reviewer_6xvT · 2022-10-21
**Interesting problem, but paper is not well-written and formulation is unclear**

**Overall Rating:** 5

**Summary:**

The authors study the problem of classification under fairness criteria in a semi-private setting, i.e., a small part of the data is clean, while a large chuck of the data's sensitive attributes are private.  They propose a framework, termed as FairSP, where the idea is to learn to "correct" the private sensitive attributes by utilizing the features of the clean data. Simultaneously, a fair classifier is learnt in an adversarial way. In the course of developing the framework, they also study the connection between privacy and fairness and note that in debiasing models, stronger privacy guarantee deteriorates fairness while in vanilla models which aren't based on debiasing, tighter privacy in fact improves fairness. The experimental evaluation of FairSP is performed on Adult, COMPAS and MEPS datasets, demonstrating improvement in fairness notions such as DP and EO over other methods.

**Strengths:**

The setting of fair classification under semi-privacy is interesting and novel. The authors make observations on the connections between fairness and privacy and justify the need to reduce noise in the private features before employing them for debiasing. They propose a framework that "corrects" private data, and design an encoder layer to learn embeddings which are utilized by functions based on adversarial debiasing.

**Weaknesses:**

The paper is lacking in these aspects:

1. The idea of adversarial debiasing has been explored by several prior works, and the primary novel idea in this work is to "correct" the private sensitive attributes and then employ them for debiasing. However, I find that the sections central to describing this (like section 4.2) are not clear, specifically in notation and definition. Are the datasets $\mathcal{D}_c$ and $\mathcal{D}_p$ corresponding to same samples? Or are they representing the clean and private part of the dataset, thus consisting of different data samples? Could you explain the insight and meaning behind equation (4)?

2. In section 4.1, do equations (1) and (2) represent $\mathcal{L}_c$ and $\mathcal{L}_p$, and not the minmax versions? Why are the loss functions constructed this way?

3. The paper is not well-written. The organization of the paper does not flow well. Some key details are pushed to the appendix. For example, it is unclear how fairness is promoted through the formulation of the method, and there is no explicit terms and neither is a an effort made to describe this. The reader has to refer to the appendix (Theorem 1) to infer this.

4. For the kind of datasets considered, it is possible that other SOTA models for fair classification which aren’t based on debiasing could have good fairness metrics. Regarding the observation in line 106 in section 2.2, could one use such models which aren't based on debiasing and improve both privacy and fairness?

5. What happens if we apply some SOTA fairness algorithms (which may not be based on debiasing or leveraging sensitive attributes) on semi-private data? A comparison would be interesting and relevant. The improvement in fairness demonstrated may not be sufficient to establish the effectiveness of FairSP.

**Overall Recommendation:**

In my opinion, this paper fails cross the acceptance threshold. Although the framework is interesting, the description of the proposed method, along with the elucidation of ideas is unclear. The notations, description and organization of the material could be better.

**Review Confidence:**

4: The reviewer is confident but not absolutely certain that the evaluation is correct

---

### Decision · Program_Chairs · 2022-10-23

Accept